# The Effect of a Ketogenic Diet versus Mediterranean Diet on Clinical and Biochemical Markers of Inflammation in Patients with Obesity and Psoriatic Arthritis: A Randomized Crossover Trial

**DOI:** 10.3390/ijms25052475

**Published:** 2024-02-20

**Authors:** Vaia Lambadiari, Pelagia Katsimbri, Aikaterini Kountouri, Emmanouil Korakas, Argyro Papathanasi, Eirini Maratou, George Pavlidis, Loukia Pliouta, Ignatios Ikonomidis, Sofia Malisova, Dionysios Vlachos, Evangelia Papadavid

**Affiliations:** 1Second Department of Internal Medicine, Attikon University Hospital, Medical School, National and Kapodistrian University of Athens, 12462 Athens, Greece; akoun@med.uoa.gr (A.K.); mankor@med.uoa.gr (E.K.); plioutaloukia@gmail.com (L.P.); 2Rheumatology and Clinical Immunology Unit, Fourth Department of Internal Medicine, Attikon Hospital, Medical School, National and Kapodistrian University of Athens, 12462 Athens, Greece; pelkats@gmail.com; 3Second Department of Dermatology and Venereology, University of Athens Medical School, 12462 Athens, Greece; argyp92@gmail.com (A.P.);; 4Department of Clinical Biochemistry, Medical School, National and Kapodistrian University of Athens, 15772 Athens, Greece; maratueirini@gmail.com; 5Second Cardiology Department, Attikon University Hospital, Medical School, National and Kapodistrian University of Athens, 12462 Athens, Greece; geopavi80@gmail.com (G.P.); ignoik@gmail.com (I.I.); 6Independent Researcher, 11142 Athens, Greece; smalisova@eviviosmed.gr; 7Independent Researcher, 16451 Athens, Greece; dvlachos@eviviosmed.gr

**Keywords:** psoriasis, psoriatic arthritis, obesity, diet intervention, Mediterranean diet, Ketogenic diet, PASI score, DAPSA score, inflammation, interleukins

## Abstract

The effect of different diet patterns on psoriasis (PSO) and psoriatic arthritis (PSA) is unknown. Τhe aim of our study was to evaluate the effectiveness of a Mediterranean diet (MD) and Ketogenic diet (KD), in patients with PSO and PSA. Twenty-six patients were randomly assigned to start either with MD or KD for a period of 8 weeks. After a 6-week washout interval, the two groups were crossed over to the other type of diet for 8 weeks. At the end of this study, MD and KD resulted in significant reduction in weight (*p* = 0.002, *p <* 0.001, respectively), in BMI (*p* = 0.006, *p <* 0.001, respectively), in waist circumference (WC) (*p* = 0.001, *p <* 0.001, respectively), in total fat mass (*p* = 0.007, *p <* 0.001, respectively), and in visceral fat (*p* = 0.01, *p <* 0.001, respectively), in comparison with baseline. After KD, patients displayed a significant reduction in the Psoriasis Area and Severity Index (PASI) (*p* = 0.04), Disease Activity Index of Psoriatic Arthritis (DAPSA) (*p* = 0.004), interleukin (IL)-6 (*p* = 0.047), IL-17 (*p* = 0.042), and IL-23 (*p* = 0.037), whereas no significant differences were observed in these markers after MD (*p* > 0.05), compared to baseline. The 22-week MD–KD diet program in patients with PSO and PSA led to beneficial results in markers of inflammation and disease activity, which were mainly attributed to KD.

## 1. Introduction

Psoriasis is one of the most prevalent auto-inflammatory diseases worldwide, with an incidence of 1.9 to 3.4% in western Europe [1]. Its etiology is considered multifactorial, and it is characterized by the dysregulation of the innate and adaptive immune systems, with the activation of T helper (Th)-1 and Th-17 T cells, leading to an increased production of inflammatory cytokines such as interleukin (IL)-1, IL-6, IL-23, IL-22, IL-17, and IL-33; tumor necrosis factor alpha (TNF-α); and interferon-gamma (IFN-γ) [2,3]. In this cascade, inflammation plays a prominent role by promoting hyper-proliferation and angiogenesis, leading to the typical skin lesions and the articular involvement of psoriatic arthritis [4].

A growing number of studies have highlighted the association between obesity and psoriasis [5,6,7,8,9]. The prevalence and incidence of psoriasis is higher among patients with obesity, while obesity is an important predisposing factor for psoriasis onset, progression, and severity. Moreover, obesity exerts a negative impact on the treatment of psoriasis and increases the adverse effect of anti-psoriatic drugs [10,11,12,13,14]. As obesity and psoriasis represent chronic inflammatory states, many recent studies have focused on the complicated role of visceral fat, which releases a number of pro-inflammatory cytokines such as TNF-α, IL-1, IL-6, and IL-8 [15]. Adipocytokines such as leptin and resistin, which are secreted not only by adipocytes but also by macrophages in adipose tissue, contribute equally to the inflammation process [16].

According to research data, lifestyle interventions, including diet, weight loss, and physical activity, not only improve the symptoms of pre-existing psoriasis, but also constitute preventable factors for the disease in individuals who are overweight and obese [17,18,19,20]. Multiple dietary patterns such as a low-calorie diet, gluten-free diet, very-low-calorie KD (VLCKD), and Mediterranean diet (MD) have been proposed for weight loss management in patients with psoriasis [21]. MD is characterized by a high consumption of fruits, vegetables, and cereals; moderate consumption of fish, olive oil, nuts, and legumes; and low intake of poultry, eggs, and red meat [22]. Recent studies have highlighted the positive effect of this eating pattern in many cardiometabolic and auto-inflammatory disorders possibly due to the reduction in oxidative stress and inflammation [23]. However, interventional randomized clinical trials to confirm these results are still lacking and the existing data rely on observational reports.

Apart from its established benefits on metabolism, KD has been considered as an alternative treatment choice for autoimmune disorders [24,25,26]. The classicKD, very-low-carbohydrate KD, Atkins diet, high-fat KD, and VLCKDare different forms of KD. The main characteristic of KD is the low content of carbohydrates (less than 30–50 g/day) along with an increase in protein and fat (75–80% kcal from fat, 5–10% kcal from carbohydrates, and 15–25% kcal from protein of the total daily energy consumption) [24]. This results in a metabolism switch to fat consumption as a main source of energy, leading to an increase in fatty acids and ketone bodies, which have anti-inflammatory properties by reducing IL-β and TNF-α plasma levels [27,28]. Based on this pathophysiological background, an increased interest regarding the effect of VLCKD on psoriasis exists, yet the evidence is still scarce. According to case reports and interventional studies, VLCKDs have been associated with a substantial improvement in the course of the disease [29,30,31]. However, randomized studies are needed to confirm the results.

The aim of our study is to compare the effectiveness of classic MD with the isocaloric KD, each applied for 8 weeks, in patients with obesity and psoriasis along with psoriatic arthritis. Our objective is to investigate the hypothesis that beyond the weight loss, the overall management of inflammation, achieved with KD for a short period of time, may lead to faster beneficial results for patients with obesity and psoriasis.

## 2. Results

### 2.1. Study Participants

In total, twenty-six patients met the inclusion/exclusion criteria of this study and were randomly assigned to KD (*n* = 13) or to MD (*n* = 13). Overall, one patient dropped out during the KD intervention, four patients during MD, and five patients during the washout period. In total, sixteen patients completed this study and were included in the final analysis (Figure 1).

The mean ± SD age of patients included in this study was 52.93 ± 7.33 years and 12 patients were female (75%). Patients included in this study displayed obesity grade II with a mean weight of 108.44 ± 19.01 and a mean body mass index (BMI) of 39.90 ± 7.60 along with increased waist circumference (WC) (122.96 ± 17.87). At baseline, patients presented moderate psoriasis with a mean Psoriasis Area and Severity Index (PASI) of 5.09 ± 5.73 along with severe PSA with a mean Disease Activity Index for Psoriatic Arthritis (DAPSA) of 46.28 ± 34.89.

### 2.2. Dietary Effects on Weight and Body Composition

At the end of this study, MD and KD resulted in significant reduction in weight (*p* = 0.002, *p* < 0.001, respectively) and BMI (*p* = 0.006, *p* < 0.001, respectively). Weight and BMI reductions were greater after the KD intervention (−10.63 kg) compared to MD (−7.48 kg), but not at a statistically significant level (*p* = 0.168, *p* = 0.200, respectively). MD and KD resulted in a significant decrease in WC (*p* = 0.001, *p* < 0.001, respectively). Patients after KD displayed a 2-fold higher reduction in WC compared to MD (−6.32% vs. −3.65%, *p* = 0.04). MD and KD resulted in significant reduction in total fat mass (*p* = 0.007, *p* < 0.001, respectively) and visceral fat (*p* = 0.01, *p* < 0.001, respectively), in comparison with baseline (Table 1).

### 2.3. Biochemical Parameters

Patients did not display any significant alteration in aminotransferases, gamma-glutamyl transferase (γ-GT), and alkaline phosphatase (ALP) (*p* > 0.05 for all parameters). KD resulted in significant reduction in serum glutamic pyruvic transaminase (SGPT) (*p* = 0.032) compared to baseline, while no significant change in SGPT was observed after MD (*p* > 0.05), in comparison with baseline. Any of the examined diet patterns resulted in significant reduction in total cholesterol (TChol), low-density lipoprotein cholesterol (LDL), high-density lipoprotein cholesterol (HDL) (*p* > 0.05 for all parameters). However, patients after KD displayed a substantial reduction in triglycerides (TGs) (*p* = 0.022) in comparison with baseline, showing an almost 3-fold higher reduction compared to MD (−11.26% versus −29.47%). No significant change in TGs was observed after MD (*p* > 0.05) (Table 1).

### 2.4. Clinical Markers of Disease Activity

After KD, patients displayed a significant reduction in PASI (*p* = 0.04) compared to baseline, showing a 2-fold higher reduction compared to an MD diet (−61.58% vs. −31.24%, *p* = 0.038). No significant change in PASI was observed after MD (*p* = 0.278). KD resulted in significant reduction in DAPSA (*p* = 0.004) compared to baseline, showing a 3-fold higher reduction compared to an MD diet (−98.62% vs. −32.64%, *p* = 0.034). No significant change in DAPSA was observed after MD (*p* = 0.06) (Table 2 and Figure 2).

### 2.5. Biochemical Markers of Inflammation

After KD, patients displayed a significant reduction in IL-6 (*p* = 0.047) compared to baseline, showing a 2-fold higher reduction compared to MD (−55.6% vs. −20.56%, *p* = 0.041). No significant change in IL-6 was observed after MD (*p* = 0.666), compared to baseline. KD resulted in substantial reduction in IL-17 (*p* = 0.042) and IL-23 (*p* = 0.037), whereas no significant differences were observed in the above markers in MD (*p* > 0.05; Table 3) (Figure 3), compared to baseline. No change in IL-22 was observed in all patients (*p* > 0.05) (Table 3).

## 3. Discussion

According to our results, the combined 22-week MD–KD program for patients with PSO and PSA on stable anti-psoriatic pharmacological treatment led to weight loss and enhancements in clinical and biochemical markers of inflammation. However, these favorable effects were mainly attributed to KD, as statistically significant reductions in clinical scores of disease activity and inflammatory markers were observed only in this dietary regimen. This study supports the correlation between weight loss and reduced disease activity and the beneficial effect of KD on the inflammatory burden of psoriasis. To our knowledge, this is the first randomized study that examined the effects of MD and KD on disease parameters and markers of inflammation in patients with PSO and PSA.

Regarding the effect of KD on patients with psoriasis, literature data are still scarce. The first relevant publication was a case report by Castaldo et al. about a 40-year-old female patient with recurrent moderate-to-severe plaque psoriasis, psoriatic arthritis, and metabolic syndrome, who received conventional treatment with adalimumab. After a disease flare, VLCKD was introduced, and led to a significant reduction in a PASI score > 80% along with the resolution of psoriatic arthralgia [29]. These favorable results were reproduced in a study that included drug-naive patients, where the intervention consisted of a 4-week and a 6-week hypocaloric, low-glycemic-index, Mediterranean-like diet. Similar to our results, a body weight reduction of 12% and a significant reduction in the PASI score (mean change of −10.6) [31] were indicated. Regarding inflammatory markers, a low-calorie KD in 30 patients with psoriasis led to decreased IL-1β and IL-2 levels, along with 10% weight loss and 50% reduction in the PASI score [30]. The pathophysiological pathways through which KD ameliorates the pro-inflammatory milieu in psoriasis have not yet been fully elucidated. The KD improves oxidative stress via the activation of nuclear factor erythroid-derived 2 (NF-E2)—related factor 2 (Nrf2), which in turn promotes the activation of the peroxisome proliferator-activated receptor-gamma (PPAR-γ) and which may elicit anti-inflammatory effects through the inhibition of nuclear factor kappa B (NF-Kb) activation [32,33]. Furthermore, β-hydroxybutyrate inhibits the NLRP3 inflammasome in lipopolysaccharide (LPS)-stimulated human monocytes, resulting in the reduction in IL-1β and IL-18 [34]. A further mechanism explaining the protective activity of the KD against oxidative stress is the intracellular modulation of the NAD^+^/NADH ratio. An increased NAD^+^/NADH ratio protects against reactive oxygen species (ROS) and plays an important role in cellular respiration, mitochondrial biogenesis, and redox reactions [35].

The implementation of a Mediterranean dietary pattern has also exhibited beneficial results in terms of autoimmunity. Skoldstam et al. [36] reported that patients with rheumatoid arthritis after MD displayed a significant reduction in the disease activity score (DAS28), and comparable results were shown in a recent study by Vadel et al. (ADIRA trial), which demonstrated the positive effect of a Mediterranean-like, anti-inflammatory diet on disease activity in patients with rheumatoid arthritis [37]. Regarding the association with psoriasis and MD, a number of studies are also available [38,39,40]. What needs to be pointed out, however, is that these studies retrospectively evaluated the adherence to MD through the use of validated questionnaires, which inevitably lack the robustness of data compared to interventional studies. For instance, a case–control study by Barrea et al. was the first that showed that patients with psoriasis had a lower adherence to MD (assessed by PREDIMED questionnaire) compared to the control group [39]. Even more importantly, psoriasis severity (assessed by PASI score and C-reactive protein (CRP)) was negatively associated with the intake of extra-virgin olive oil, fruits, nuts, fish or seafood, vegetables, and legumes, and it was positively correlated with red meat intake [39]. In the same notion, a cross-sectional observational study with a larger sample (*n =* 35.735 patients) by Phan et al. showed an inverse association between adherence to a Mediterranean diet (assessed by MED-LITE) and psoriasis severity [39]. These beneficial effects of MD on disease activity and inflammation markers were also demonstrated in our study, despite not achieving statistical significance. The reason for this discrepancy could be the small size of the sample, along with the fact that the patients already suffered from psoriatic arthritis, which implies the significantly more severe inflammatory burden of the disease compared to the aforementioned trials. Based on these data, the Medical Board of the National Psoriasis Foundation recommends a trial of MD in patients with psoriasis, but the need for large-scale, interventional trials is still underlined [41].

The positive effects of MD cannot be attributed specifically to its every single component separately, but it is rather the combination of all the different macro- and micronutrients, which exert the favorable anti-inflammatory effects. However, extra-virgin olive oil seems to play the most important role [42]. Oleic acid, a monounsaturated fatty acid (MUFA), is its major constituent, and is highly effective in decreasing the oxidization of LDL [43]. In a study by Loued et al., the consumption of extra-virgin olive oil for 12 weeks significantly increased the anti-inflammatory activities of both HDL and paraoxonase 1 (PON1) and decreased serum levels of inter cellular adhesion molecule (ICAM-1), which is one of the most prominent adhesion molecules with pro-inflammatory properties. Olive oil down-regulates pro-inflammatory cytokines such as IL-6 and TNF-α [42].

The present study has several strengths. First, we employed a crossover design to reduce the effects of inter-individual variation and maximize the statistical power from the available sample size. We also implemented a 6-week washout period, which we believe to be sufficient to normalize effects from the prior dietary period. In addition, it is the first study that is interventional in nature and performs a direct comparison between these two dietary patterns.

One limitation of our study is its small sample. Furthermore, our results could not easily be extrapolated to a real-world setting as the patients were evaluated at bi-weekly intervals, which ensured a high degree of adherence but it is not easily feasible in every-day clinical practice. Nevertheless, from a clinical point of view, the beneficial effect of diet interventions as an adjunct treatment to conventional pharmacological therapy was confirmed, and these two dietary regimens could be implemented alternately in patients with psoriasis.

The present study demonstrated that the combined 22-week Mediterranean–Ketogenic diet program for patients with PSO and PSA led to beneficial results in indices of disease activity and pro-inflammatory markers. These favorable effects were mainly attributed to KD, but MD also showed a beneficial tendency. These findings further establish the association between dietary interventions and auto-inflammatory disorders and emphasize the need for more, large-scale interventional trials to compare different dietary patterns.

## 4. Materials and Methods

### 4.1. Study Population

Sixteen patients who were admitted to the dermatology unit with the diagnosis of psoriasis and psoriatic arthritis were enrolled in this study. Inclusion criteria were the following: (1) age above 18 years, (2) BMI ≥ 30 kg/m^2^, (3) diagnosis of psoriasis and psoriatic arthritis, (4) constant systematic treatment with biologic agents and/or synthetic-disease-modifying anti-rheumatic drugs (DMARDs) for at least 3 months, (5) PASI score improvement < 75%, with moderate or severe activity in joints (>3 swollen and >3 tender joints or DAPSA < 14). Exclusion criteria were the following: participation in another study, eGFR < 60 mL/min/1.73 m^2^, malignancy, severe hepatic disorder, HbA1C > 10%, and use of glucagon-like-petide-1 analogues.

### 4.2. Study Design

This is a randomized, open-label, controlled crossover study conducted from May 2020 to January 2022 at the Unit of Diabetes, Second Department of Internal Medicine, in collaboration with the Dermatology Department in Attikon University Hospital. Clinical and medical measurements, as well as lifestyle intervention sessions, were performed at the Attikon University Hospital. The patients were randomly assigned to two groups, starting either with MD or KD for a period of 8 weeks. After a 6-week washout interval, the two groups were crossed over to the other type of diet for the same 8-week period (Figure 4).

Basic characteristics, including age, gender, smoking status, alcohol consumption, physical activity level, medical history, concomitant medication, and dietary habits, were obtained at baseline. All participants underwent a physical examination during the first visit. Before and after each intervention, the study team performed anthropometric measurements, assessment of the PASI and DAPSA score, and blood sampling for biochemical markers of inflammation and hormones.

The study protocol was approved by the ethics committee of Attikon University Hospital (171/09-04-2020) before the beginning of the enrollment and any other procedure. All participants provided written informed consent before study initiation. This study was carried out in accordance with the Declaration of Helsinki (Trial Registration: Clinicaltrials.gov, accessed on 13 January 2024; Identifier: NCT06164860).

### 4.3. Anthropometric Measurements

Anthropometric parameters were obtained at the beginning and at the end of each intervention. Height was measured using a stadiometer and weight was measured using a calibrated electronic scale. Body composition was assessed by a Tanita BC-420 body composition analyzer. Waist and hip circumference were taken at a themed point between the lower rib margin and the iliac crest and at the widest part of the hips, respectively, with a stretch-resistant tape kept parallel to the ground. Study subjects had their waist uncovered and were asked to stand with their feet close together and their weight equally distributed on each leg. 

### 4.4. Biochemical Measurements

Metabolic parameters were obtained before and after each intervention. The biochemical parameters that were evaluated included glucose, TGs, TChol, HDL, LDL, aminotransferases, alkaline phosphatase (ALP). Moreover, plasma levels of IL-6, IL-17, IL-23, and IL-22 were also evaluated by ELISA kits. Plasma samples were obtained by venipuncture and stored at −80 °C. Concentrations of interleukins were measured using a solid-phase enzyme-linked immunosorbent assay (ELISA) following manufacturer’s instructions. Plasma levels of IL23, IL17A, and IL6 were determined using commercially available ELISA kits (Mabtech AB, Nacka Strand, Sweden) (sensitivity, % intra- and inter-precision was 5 pg/mL, 1 pg/mL, and 2 pg/mL, respectively, and <10%) and levels of IL22 (OriGene Technologies, Inc., Rockville, MD, USA) (15 pg/mL sensitivity, % intra- and inter-precision < 10%).

### 4.5. Assessment of Psoriasis and Psoriatic Arthritis Severity

The severity of the disease was rated using the PASI score [38], which accounts for the extent of psoriatic involvement of the body surface areas on the head, trunk, arms, and legs, in addition to the severity of scale formation, erythema, and plaque indurations on each region of the body (score range: 0–72; higher scores indicate more severe disease). The severity of psoriatic arthritis was assessed by using DAPSA. The DAPSA score was calculated by adding the number of tender and swollen joints, visual analogue scale (VAS) pain, patient’s global assessment (PtGA), and CRP (mg/dL).

### 4.6. Dietary Interventions

Both dietary interventions were designed to provide approximately 1550 (±50) calories per day. The KD provided approximately 34% proteins, 55% fat, and 11% carbohydrates. The macronutrient targets were successfully met by replacing breakfast and two daily snacks with products with the best nutritional value/macronutrient content for a Ketogenic diet, provided by Evivios Med (Athens, Greece). Lunch and dinner were natural protein-rich dishes. Each participant received accurate teaching and a brochure that incorporated a template of a weekly diet plan from an expert dietitian. More specifically, the brochure contained a daily meal plan (each meal had options they could choose from), a portion guide for foods containing protein, a list with the allowed vegetables (based on their carbohydrate content) that could be used in the salad alternatives for olive oil, allowed foods (without calories) such as sweeteners and spices, and other general information. It was proposed that the patients consume 5 meals per day. A typical example of such a meal would be a 120 g oven-baked chicken, 2 cups of salad (only with the allowed vegetables), and 1 tablespoon of olive oil.

The MD provided 20% proteins, 40% fat, and 40% carbohydrates. Each participant received a brochure with a template of a daily five-course (breakfast, lunch, dinner, and two snacks) diet plan that provided several options for each meal and were advised to use it as a guide for the 8-week period. It was proposed that the patients consume 5 meals per day. A typical example of such a meal would bea 90 g oven-baked chicken, salad with 2 teaspoons of olive oil, 1 cup of boiled rice.

The participants met the same expert dietitian every two weeks during the entire study period. They were encouraged to keep a food diary, which assisted them to monitor food consumption and promoted participants’ awareness regarding their adherence.

Participants who missed a scheduled visit were advised to attend a make-up counseling session the following week.

Every 2 weeks, participants were asked to attend a face-to-face (F2F) dietetic consultation in an outpatient’s clinic. During each follow-up consultation, the expert dietitian assessed participants’ food diary and tailored offered advice that aimed to encourage participants’ adherence to the dietary intervention they were assigned to.

### 4.7. Primary and Secondary Endpoints

The primary endpoint was changes in clinical scores of the disease activity (PASI, DAPSA), 8 weeks after KD compared to the 8-week MD diet intervention. Secondary endpoints were biochemical markers of inflammation (IL-6, IL-17, IL-22, IL-23) and alterations in anthropometric parameters 8 weeks after KD compared to the 8-week MD diet intervention.

### 4.8. Statistical Analysis

The sample size was estimated based on the difference in the DAPSA score after the intervention. DAPSA score improvement ≥ 50% was considered as clinically significant [44]. To be able to detect a difference of ≥50% in the DAPSA score with 80% power at a 5% significance level with a two-sided paired *t*-test, 15 patients should complete this study. To account for possible dropouts, a total of 26 patients were enrolled. The statistical analysis was performed using the SPSS 22.0 statistical software package (SPSS Inc., Chicago, IL, USA). All variables are expressed as the mean ± SD. Categorical variables are expressed as percentages of the population. Continuous variables were tested by the Kolmogorov–Smirnov test to assess the normality of distribution. Variables with a non-normal distribution were analyzed after transformation into ranks. Categorical data were analyzed using the χ^2^ test. All statistical tests were 2-tailed, and *p* < 0.05 was considered to be the level of statistical significance.

## 5. Conclusions

In our study, we showed that the combined 22-week Mediterranean–Ketogenic diet program for patients with PSO and PSA led to beneficial results in indices of disease activity and pro-inflammatory markers. These beneficial changes were mainly attributed to KD, but MD also showed a beneficial tendency. Although the effects of MD and KD are well established in a range of cardiometabolic diseases, their possible efficacy as an adjunct treatment to conventional pharmacological therapy in patients with psoriasis is a novel clinical implication. These findings further establish the association between dietary interventions and auto-inflammatory disorders and emphasize the need for more, large-scale interventional trials to compare different dietary patterns.

## Figures and Tables

**Figure 1 ijms-25-02475-f001:**
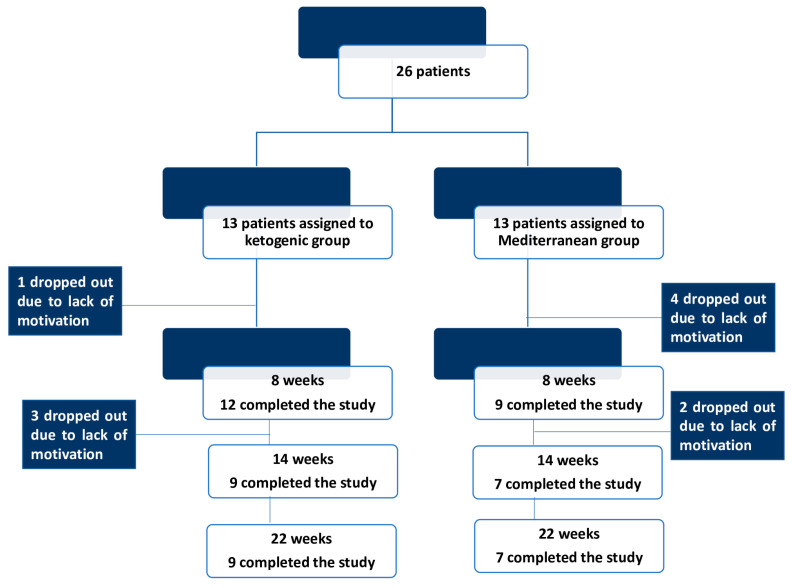
CONSORT flow diagram showing the progress through the phases of the trial.

**Figure 2 ijms-25-02475-f002:**
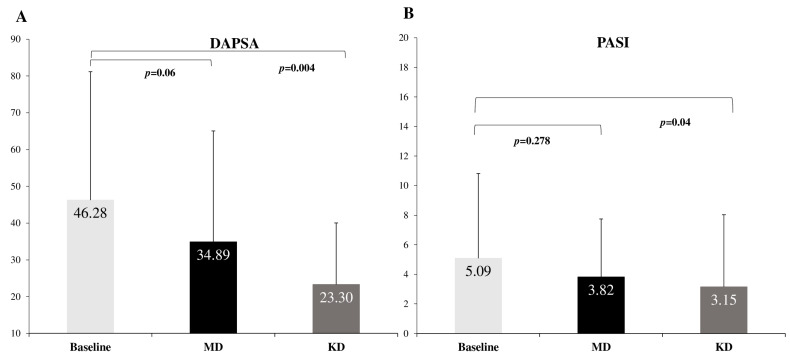
The effect of diet intervention on (**A**) DAPSA and (**B**) PASI. MD: Mediterranean Diet; KD: Ketogenic Diet; PASI: Psoriasis Area and Severity Index; DAPSA: Disease Activity Index for Psoriatic Arthritis.

**Figure 3 ijms-25-02475-f003:**
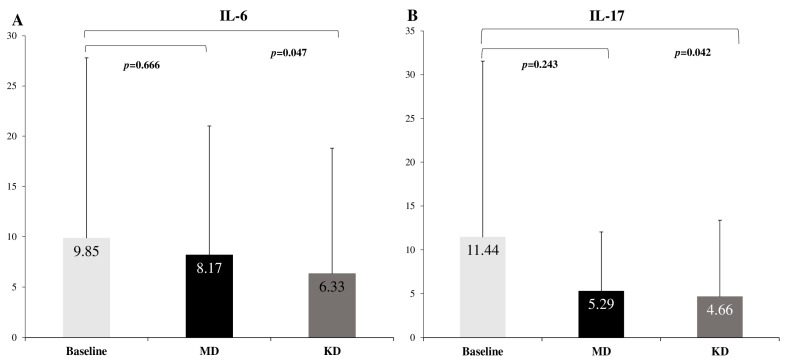
The effect of diet intervention on (**A**) ΙL-6 and (**B**) IL-17. MD: Mediterranean diet; KD: Ketogenic diet.

**Figure 4 ijms-25-02475-f004:**
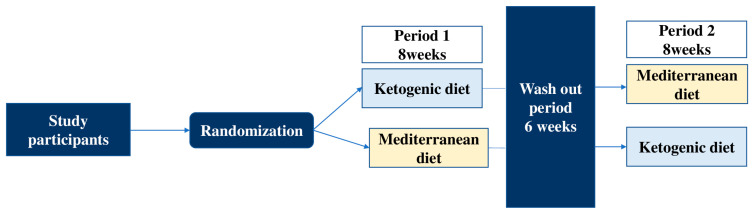
Design and analysis of crossover trial.

**Table 1 ijms-25-02475-t001:** Τhe effect of the diet intervention on anthropometric measurements and biochemical parameters.

	Baseline	MD	Δ%	KD	Δ%	*p*-Value ^†^	*p*-Value *	*p*-Value ^⁋^
Weight (kg)	108.44 ± 19.01	101.21 ± 17.95	−7.48	98.17 ± 17.46	−10.63	0.002	<0.001	0.168
BMI (kg/m^2^)	39.90 ± 7.60	37.40 ± 7.75	−7.28	36.30 ± 7.33	−10.18	0.006	<0.001	0.200
WC (cm)	122.96 ± 17.87	118.53 ± 15.71	−3.65	115.56 ± 15.52	−6.32	0.001	<0.001	0.040
Fat Mass (kg)	46.83 ± 12.76	42.30 ± 12.84	−12.45	40.51 ± 12.80	−17.47	0.007	<0.001	0.320
Visceral Fat (%)	15.43 ± 4.38	14.18 ± 4.38	−10.79	13.50 ± 3.81	−15.20	0.01	<0.001	0.326
Glu (mg/dL)	97.28 ± 21.70	98.21 ± 32.30	−2.90	103.62 ± 23.56	4.82	0.927	0.403	0.436
SGOT (IU/L)	19.14 ± 8.82	16.76 ± 5.67	−18.79	20.52 ± 13.45	−9.57	0.285	0.759	0.574
SGPT (IU/L)	18.57 ± 11.21	15.56 ± 11.46	−23.70	14.01 ± 7.84	−25.94	0.283	0.032	0.905
γ-GT (mg/dL)	24.40 ± 17.65	24.20 ± 20.00	−13.01	18.60 ± 12.96	−32.84	0.958	0.022	0.142
ALP (IU/L)	67.5 ± 25.72	59.83 ± 14.25	−17.99	57.75 ± 15.35	−19.76	0.371	0.127	0.935
TChol (mg/dL)	181.84 ± 45.19	185.74 ± 36.04	2.5	174.95 ± 36.24	−5.3	0.557	0.496	0.292
HDL (mg/dL)	44.57 ± 11.25	41.55 ± 7.97	−8.21	43.96 ± 7.05	−1.04	0.387	0.800	0.439
LDL (mg/dL)	110.99 ± 53.11	96.95 ± 23.02	−17.5	87.60 ± 21.23	−32.1	0.256	0.164	0.547
TGs (mg/dL)	133.63 ± 40.61	127.82 ± 47.95	−11.26	106.90 ± 29.24	−29.47	0.632	0.022	0.146

Data are presented as mean ± SD. Δ% indicates percentage change from baseline. MD: Mediterranean diet; KD: Ketogenic diet; BMI: Body mass index; WC: Waist circumference; Glu: Glucose; TGs: Triglycerides; TChol: Total cholesterol; HDL: High-density lipoprotein cholesterol; LDL: Low-density lipoprotein cholesterol; SGOT: Serum glutamic-oxaloacetic transaminase; SGPT: Serum glutamic pyruvic transaminase; γ-GT: Gamma-glutamyl transferase; ALP: Alkaline phosphatase. ^†^ Comparisons between baseline and after MD, * comparisons between baseline and after KD, ^⁋^ comparisons between MD and KD.

**Table 2 ijms-25-02475-t002:** Τhe effect of the diet intervention on clinical markers of disease activity.

	Baseline	MD	Δ%	KD	Δ%	*p*-Value ^†^	*p*-Value *	*p*-Value ^⁋^
PASI	5.09 ± 5.73	3.82 ± 3.93	−33.24	3.15 ± 4.88	−61.58	0.278	0.040	0.038
DAPSA	46.28 ± 34.89	34.89 ± 30.17	−32.64	23.30 ± 16.75	−98.62	0.060	0.004	0.034

Data are presented as mean ± SD. Δ% indicates percentage change from baseline. MD: Mediterranean diet; KD: Ketogenic diet; PASI: Psoriasis Area and Severity Index; DAPSA: Disease Activity Index for Psoriatic Arthritis. ^†^ Comparisons between baseline and after MD, * comparisons between baseline and after KD, ^⁋^ comparisons between MD and KD.

**Table 3 ijms-25-02475-t003:** Τhe effect of the diet intervention on biochemical markers of inflammation.

	Baseline	MD	Δ%	KD	Δ%	*p*-Value ^†^	*p*-Value *	*p*-Value ^⁋^
IL-6	9.85 ± 17.94	8.17 ± 12.85	−20.56	6.33 ± 12.47	−55.6	0.666	0.047	0.041
IL-17	11.44 ± 20.10	5.29 ± 6.74	−116.25	4.66 ± 8.72	−145.49	0.243	0.042	0.687
IL-23	23.59 ± 11.04	19.15 ± 9.70	−23.18	17.86 ± 9.97	−32.08	0.151	0.037	0.540
IL-22	190.24 ± 166.10	213.43 ± 211.86	10.79	240.87 ± 256.35	20.83	0.584	0.328	0.368

Data are presented as mean ± SD. Δ% indicates percentage change from baseline. MD: Mediterranean diet; KD: Ketogenic diet. ^†^ Comparisons between baseline and after MD, * comparisons between baseline and after KD, ^⁋^ comparisons between MD and KD.

## Data Availability

Study data are available upon reasonable request.

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
