# Peer review of "The Effect of a Ketogenic Diet versus Mediterranean Diet on Clinical and Biochemical Markers of Inflammation in Patients with Obesity and Psoriatic Arthritis: A Randomized Crossover Trial"

_ijms, 2024, doi:10.3390/ijms25052475_

Round 1

Reviewer 1 Report

Comments and Suggestions for Authors

Dear Authors,

My main suggestions are included in the file below.

Comments on the Quality of English Language

In my opinion, a review by a native speaker would be beneficial.

Author Response

Thank you for your invaluable comments.

1. All grammar and spelling corrections (1-46) have been adressed.

2. All the comments regarding the abbrevations  (1-14) have been adressed.

3. The figures have been reformatted according to your suggestions.

4. Please clarify the comment «It looks like Figures 3 and 4 are swapped».

5. Reformating has been performed.

6. A relevant reference (ref. 35) has been added.

7. The requested information for ELISA KIT has been added.

8.  The country of origin has been added accordingly.

9. All the requested information about diet intervention has been added. 

10. No funding was received for this study. The study was conducted with own financial resources.

Reviewer 2 Report

Comments and Suggestions for Authors

To the Authors

Brief Summary

The objective of this 8-week, 2-arm randomized controlled trial was to compare the effectiveness of the Mediterranean diet (MD) and Ketogenic diet (KD), in 16 patients suffering from Psoriasis. Data analysis showed that by the end of the 8-week dietary intervention, significant reductions in weight and adiposity (as measured by BMI, WC, total fat mass and visceral fat), were observed in both groups. Regarding biochemical parameters, non-significant improvements were observed in the lipid profiles of patients in both groups. Notably, statistical significance was observed in the KD group only for TG, PASI, DAPSA, as well as in inflammatory markers. Overall, compared to the MD group, greater reductions were observed in the KD group. The investigators concluded that this clinical trial demonstrated the anti-inflammatory potential of MD and KD in adult patients with Psoriasis. Nonetheless, the observed beneficial effects were greater for those adhering to the KD. Worth consideration is the efficacy of both diets in promising successful weight loss in these patients.

Indeed, the burden of Psoriasis in the adult population is unequivocal, and more research is urgently needed to identify non-pharmacological and non-invasive approaches that could relieve patient burden.

Overall, the manuscript was well-written. However, there are a couple minor points that require clarification. Please refer to my comments to the authors below. We look forward to more studies from this research group. Good luck with future publications.

Comments to the authors

Abstract

-Line 32. Please define PASI and DAPSI in text

Main manuscript

-Line 64. Define MD the first time that it appears in text.

Study design

-On what criteria was the intervention time period of 8-weeks chosen?

-Did you estimate sample size calculation?

-Mention the minimum difference or effect that represents clinical importance?

Statistical analysis:

-I am not a statistical expert, could you explain how you calculated the %∆ from baseline, using which mathematical formula.

Results

-Line 143 ‘KD resulted in 142 significant reduction in DAPSA (p=0.04) compared to baseline.’

According to Table 2, the P-value for DAPSA should be P = 0.004 and not 0.04.

-Line 144 ‘No significant change of DAPSA was observed after MD (P = 0.073), (table 2), (fig.2).

Likewise, regarding MD, shouldn’t the P-value for DAPSA be P=0.06 and not 0.073?

Discussion

-Are these results clinically significant?

Limitations

Please acknowledge in the limitations, one more source of bias that significant results could have been obtained by chance due to small sample size.

Author Response

Thank you for your invaluable comments.

Lines 32 and 64:corrections have been made

The intervention time period of 8weeks has been chosen based on the literature and the experience of our team as a sufficient duration to have beneficial effects and a high rate of compliance, without adverse effects.

The requested information regarding sample size calculation has been added

Δ% has been calculated as: [(post-intervention value - baseline value)/baseline value] x 100.

Regarding the results appropriate corrections have been made

Regarding the clinical significance of our results a comment after limitaton section was added

Reviewer 3 Report

Comments and Suggestions for Authors

Lambadiari et al. present an interesting report on beneficial effects of Mediterranean and Ketogenic diet regimens on a population of patients affected by obesity and psoriatic arthritis. The abstract is somewhat confusing and I suggest rewriting it in a more orderly fashion. The dietary alternation is unclear even if it is later explained in the text. Wash out, for instance, is indicated but comes a bit out of blue there.

In describing the diets, more detailed information should be provided because there are many variables, especially in the vitamin regime that could result from MD, which could have strong effects on inflammation markers.

Vitamin D and C and unsaturated fatty oils intake, at least, should hopefully be estimated. Was the intake of red meat singled out?

Were there drug contributions in these patients that have not been disclosed?

What were circadian habits? Rest times? 

Overall the conclusion is interesting but, since effects of MD and/or KD are well known in the literature, a more detailed picture could heighten the interest of this paper for readers. 

Comments on the Quality of English Language

English quite fine, some mispelling that could be fixed during copy editing 

Author Response

The abstract is somewhat confusing and I suggest rewriting it in a more orderly fashion. The dietary alternation is unclear even if it is later explained in the text. Wash out, for instance, is indicated but comes a bit out of blue there.

According to the authors guidelines word limit for the abstract is 200 words, without headings. Therefore, not many details can be given regarding the diets. The wash out interval is mentioned here because is an intergal part of the cross-over desing of the trial.

In describing the diets, more detailed information should be provided because there are many variables, especially in the vitamin regime that could result from MD, which could have strong effects on inflammation markers. Vitamin D and C and unsaturated fatty oils intake, at least, should hopefully be estimated.

None of these were estimated. The meal plans provided meal options that participants could choose from, so it was not possible to measure the exact fat and micronutrient content.

Was the intake of red meat singled out?

No, it was not.

Were there drug contributions in these patients that have not been disclosed?

The patients did not recieve any treatment related to obesity. The antipsoriatic treatment was stable for three months prior to study enrollment and during the study.

What were circadian habits? Rest times?

All the patinets presented similar dialy cicardian patterns, working from till afternoon and sleeping during the night.

Overall the conclusion is interesting but, since effects of MD and/or KD are well known in the literature, a more detailed picture could heighten the interest of this paper for readers.

A comment has been adressed in the conclusion section

Round 2

Reviewer 1 Report

Comments and Suggestions for Authors

Dear Authors,

The corrections made are satisfactory to me and this version of the manuscript is better than the previous one.